# Hongtang Bridge Expansion Joints InSAR Deformation Monitoring with Advanced Phase Unwrapping and Mixed Total Least Squares in Fuzhou China

**DOI:** 10.3390/s25010144

**Published:** 2024-12-29

**Authors:** Baohang Wang, Wu Zhu, Chaoying Zhao, Bojie Yan, Xiaojie Liu, Guangrong Li, Wenhong Li, Liye Yang

**Affiliations:** 1School of Geography and Oceanography, Minjiang University, Fuzhou 350108, China; wangbaohang@mju.edu.cn (B.W.); bnunercita@163.com (B.Y.); 2Key Laboratory of Ecological Geology and Disaster Prevention, Ministry of Natural Resources, Xi’an 710054, China; 3School of Geological Engineering and Surveying and Mapping, Chang’an University, Xi’an 710061, China; cyzhao@chd.edu.cn (C.Z.); 2020226022@chd.edu.cn (G.L.); 4School of Civil Engineering, Lanzhou University of Technology, Lanzhou 730050, China; xiaojie_liu_cd@163.com; 5Xi’an Center of Geological Survey, China Geological Survey, Xi’an 710054, China; liwenhong01@mail.cgs.gov.cn; 6College of Civil Engineering, Xiangtan University, Xiangtan 411105, China; yangliye@chd.edu.cn

**Keywords:** InSAR, bridge expansion joints, Hongtang bridge, network optimization, mixed total least squares, independent component analysis

## Abstract

Bridge expansion joints are critical components that accommodate the movement of a bridge caused by temperature fluctuations, concrete shrinkage, and vehicular loads. Analyzing the spatiotemporal deformation of these expansion joints is essential for monitoring bridge safety. This study investigates the deformation characteristics of Hongtang Bridge in Fuzhou, China, using synthetic aperture radar interferometry (InSAR). We optimize the network paths to enhance the phase unwrapping process of InSAR. Additionally, to address design matrix bias resulting from inaccurate temperature data, we employ the mixed total least squares method to estimate deformation parameters. Subsequently, we utilize independent component analysis to analyze the spatiotemporal deformation characteristics of the bridge. The average standard deviation of the unwrapped phase and the modeling residuals have been reduced by 87% and 5%, respectively. Our findings indicate that thermal expansion deformation is primarily concentrated in the expansion joints, measuring approximately 0.6 mm/°C. In contrast, the cable-stayed bridge deck exhibits the largest deformation magnitude, exceeding 2.0 mm/°C. This research focuses on bridge structures to identify typical deformation locations and evaluate their deformation characteristics. Such analysis is beneficial for conducting safety assessments of bridges.

## 1. Introduction

Bridge infrastructure, as one of the critical transportation hubs, plays a significant role in promoting social and economic development. However, factors such as the aging of structural materials, environmental degradation, and vehicle overloading contribute to the damage and deterioration of bridges. This deterioration leads to a reduction in structural bearing capacity, impacting the normal operation of bridges, and potentially resulting in casualties and property losses. Synthetic Aperture Radar (InSAR) is an image geodesy technique that can capture small deformation features of monitoring targets. InSAR technology has been widely utilized for monitoring bridge deformation [1,2,3,4,5,6,7,8], particularly with the availability of Sentinel-1A data.

Advanced temporal InSAR technologies have been developed, including permanent scatterer interferometry and its enhanced versions [9,10,11,12], small baseline subset interferometry [13], distributed scatterer interferometry utilizing spatiotemporal filtering [14,15,16,17,18], and differential tomographic SAR interferometry [19,20]. These technologies have been thoroughly analyzed and compared [21,22,23,24,25]. Given the vast amounts of SAR data, various InSAR online services [26], parallel InSAR data-processing methods [27], and dynamic data-processing techniques [28] have been proposed.

In terms of SAR interferometric bridge monitoring, Fornaro et al. utilized the TomoSAR SAR technique to analyze temperature-induced deformation patterns [29]. Crosetto et al. extended the analysis from a two-dimensional to a three-dimensional model, incorporating temperature-induced deformation, and discovered that the distribution of temperature-related deformation correlated with the structural characteristics of the bridge [30]. Huang et al. estimated the temperature-induced deformation of high-speed railway bridges using Sentinel-1 data [31]. Qin et al. identified both trend deformation and temperature-induced deformation in two arch bridges by utilizing the InSAR method [32]. Zhang et al. employed a terrestrial microwave radar interferometer known as the GAMMA Portable Radar Interferometer (GPRI) to measure the deformation characteristics of bridges [33]. Selvakumaran et al. applied the corner reflector-based InSAR method to assess Waterloo Bridge [34]. The displacements of the Hong Kong–Zhuhai–Macao Bridge (HZMB) were analyzed using Sentinel-1A observations [35,36]. Martin et al. utilized independent component analysis technology to enhance InSAR’s capability in detecting bridge deformation characteristics [37]. Song et al. implemented bridge structure segmentation to facilitate InSAR monitoring of bridge deformation [38]. Wang et al. proposed an L1 and L2 norm InSAR bridge phase unwrapping method [39]. Additionally, InSAR combined with distributed scattering interferometry [40], ray tracing [41], and GPS [42] methods has been proposed to monitor bridge deformation.

Bridge expansion joints connect different bridge decks and play a crucial role in distributing loads and accommodating temperature-induced deformations. The deformation of these joints is primarily influenced by temperature fluctuations, traffic loads, wind, and other factors, leading to misalignment, tension, and torsional deformations of the bridge during operation. The direction of temperature deformation propagation is associated with the specific geometric shape of the structure, typically extending along the longest side of that shape [43]. Moreover, the magnitude of temperature deformation is closely related to the material properties and the effective propagation length of the structure. Significant temperature changes can result in cracks, particularly in the expansion joints. These cracks may occur when temperature-induced material shrinkage or expansion exceeds the tensile strength of the material [44,45].

This study utilized 98 Sentinel-1A SAR datasets collected from 5 April 2021 to 6 July 2024, to monitor the deformation of Hongtang Bridge. We analyzed the deformation characteristics based on the structural information of the bridge and found that the deformation is segmented by expansion joints, exhibiting opposing deformation patterns. These deformation characteristics are significantly correlated with temperature variations. To ensure the reliability of the InSAR deformation monitoring results, we utilized the temporal coherence of the double-difference phase and the number of redundant observations strategy to identify reliable network arcs and obtain the unwrapped deformation phase. The daily average temperature for the study area [46], often differs from the actual temperature of the bridge during SAR data acquisition. Subsequently, we employed mixed-integer least squares inversion to analyze temperature-related deformation parameters, thereby mitigating the impact of temperature deviations on our deformation parameters. Finally, based on the results of on-site investigations, InSAR deformation data, and the Independent Component Analysis (ICA) method, we conducted an analysis of the deformation characteristics of the expansion joints for Hongtang Bridge.

## 2. Study Area SAR Dataset

The new Hongtang Bridge is situated in the western suburbs of Fuzhou City, Fujian Province, spanning the Min River. Officially opened on the evening of 21 April 2021, the bridge features an eight-lane mainline, with a total length of 2200 m and a width of 64 m. It incorporates a cable-stayed bridge deck to accommodate the shipping capacity of the Wulong River, as illustrated in Figure 1A.

To monitor the deformation characteristics of the bridge, we collected 98 Sentinel-1A SAR points from 5 April 2021 to 6 July 2024. Table 1 presents the parameters of the Sentinel-1A. Figure 1A,B illustrates the average SAR intensity image in the SAR coordinate system (range and azimuth direction) and the spatiotemporal baseline of the interferograms. The heading angle of the SAR satellite is 12.4°. The red rectangular box highlights the location of Hongtang Bridge over Wulong River. To enhance the signal-to-noise ratio, we applied multi-look filtering with a ratio of 2:1 in both the range and azimuth directions. Consequently, the resolution of SAR data in the range and azimuth directions is 4.7 × 14.0 m. Digital elevation model (DEM) data are sourced from the Shuttle Radar Topography Mission (SRTM) and have a resolution of 30 m.

## 3. Methodology

The flowchart for monitoring the deformation of bridge expansion joints using multi-temporal Sentinel-1A data is presented in Figure 2. Firstly, we generate interferograms using external DEM, orbital data, and Single-Look Complex (SLC) data. We then utilize temporal phase stability to identify the primary coherent pixels. Next, we employ K-Nearest Neighbors (KNN) technology to establish a redundant observation network, followed by ambiguity detection and redundant observation methods to select a reliable phase unwrapping network. Subsequently, we perform spatial phase unwrapping and inversion of deformation parameters using the mixed total least squares method. Finally, based on the results of on-site investigations and InSAR deformation data, we conduct an analysis of the deformation characteristics of Hongtang Bridge expansion joints using independent component analysis techniques.

### 3.1. Network Optimization Assisted InSAR Phase Unwrapping

Popular L1 norm phase unwrapping methods can estimate integer ambiguity from Delaunay network arcs [47]. The presence of excessive ambiguity and noise in the phase of network arcs can reduce the performance of phase unwrapping. We can identify reliable phase unwrapping paths from redundant networks. In contrast to the Delaunay network, a free network can generate redundant observation networks, such as the K-Nearest Neighbors (KNN) method, which connects one pixel to its nearest N pixels. Subsequently, we utilize the temporal coherence of the double-difference phase to identify reliable network arcs [9,48], as demonstrated in Equation (1).
(1)$$\begin{array}{l} \gamma^{p,q}=\frac{1}{M}\left|\sum_{ifg=1}^{M}\exp(j\Delta \varphi_{ifg}^{p,q})\right|\\ \Delta \varphi_{ifg}^{p,q}=wrap(\varphi_{ifg}^{p}-\varphi_{ifg}^{q}) \end{array}$$
where $\gamma^{p,q}$, *M*, and $\Delta \varphi_{ifg}^{p,q}$ represent the temporal coherence of arcs, number of SAR interferograms, and double-difference phase of two neighboring pixels *p* and *q*, respectively. After estimating the temporal coherence for all redundant arcs, we can utilize a threshold $\gamma_{threshold}^{p,q}$ to select reliable arcs. After that, to control the phase unwrapping network structure, the number of redundant observation strategies [49] is used to optimize network structures as shown in Equation (2).
(2)$$Pixel_{i}^{NRO}<T_{NRO}$$
where $Pixel_{i}^{NRO}$ and $T_{NRO}$ are the number of redundant observations of each pixel and set threshold. In the optimized phase unwrapping network, if a pixel-connected arc exceeds a specified threshold, we retain only $T_{NRO}$ arcs that exhibit the highest temporal coherence. Afterward, we eliminate low-quality arcs and isolated pixels. Subsequently, we apply the L1 norm phase unwrapping method to optimized networks, which is based on edge constraints derived from the optimized network [50,51].

### 3.2. Thermal Expansion Coefficient Estimation Through Mixed Total Least Squares

After obtaining the absolute deformation phase, we estimate the deformation parameters, including the digital elevation model (DEM) error, deformation rate, and thermal expansion coefficient, as shown in Equation (3),
(3)$$\begin{array}{l} \underset{A_{1}}{\underset{︸}{\left[\begin{array}{c} \begin{array}{cc} \frac{4\pi }{\lambda R\sin\vartheta }B_{1} & \frac{4\pi }{\lambda }\Delta t_{1} \end{array}\\ \begin{array}{cc} \vdots & \vdots \end{array}\\ \begin{array}{cc} \frac{4\pi }{\lambda R\sin\vartheta }B_{M} & \frac{4\pi }{\lambda }\Delta t_{M} \end{array} \end{array}\right]}}\underset{X_{1}}{\underset{︸}{\left[\begin{array}{c} \Delta H\\ V \end{array}\right]}}+\underset{A_{2}}{\underset{︸}{\left[\begin{array}{c} \frac{4\pi }{\lambda }\Delta d_{1}\\ \vdots \\ \frac{4\pi }{\lambda }\Delta d_{M} \end{array}\right]}}\underset{X_{2}}{\underset{︸}{TH}}=\underset{L}{\underset{︸}{\left[\begin{array}{c} \theta_{1}\\ \theta_{2}\\ \vdots \\ \theta_{M} \end{array}\right]}}\\ A_{1}X_{1}+A_{2}X_{2}=L \end{array}$$
where θ,λ, *R*, ϑ, *B*, Δt, Δd, and *M* represent the unwrapped phase, wavelength, slant range between radar sensor and the target, incidence angle, perpendicular baseline, interferogram time interval, temperature difference between two SAR scenes, and number of interferograms, respectively. The Δ*H*, *V,* and *TH* represent the DEM error, deformation rate, and thermal expansion coefficient, respectively.

In three-dimensional deformation estimation, Yin et al. employed the total least squares method to mitigate design matrix errors resulting from inaccurate DEM [52]. The thermal expansion component requires temperature data to construct a design matrix. Utilizing the average temperature of a region, rather than the specific temperature of the bridge, introduces bias into the design matrix A_2_. The unwrapping phase *L* usually contains errors, including atmospheric noise and decorrelation phase, etc. In contrast, the design matrix A_1_ is free from errors. Consequently, we used the mixed total least squares method [53,54] to estimate the deformation parameters. Initially, we perform QR decomposition on the coefficient matrices A_1_ to obtain the matrices $Q=\left[\begin{array}{cc} Q_{1} & Q_{2} \end{array}\right]$ and R, as shown in Equation (4).
(4)$$A_{1}=QR$$

Then, we left-multiply the model Equation (3) by the Q matrix to derive Equation (5).
(5)$$\begin{array}{l} Q^{T}AX=Q^{T}L\\ \left[\begin{array}{cc} Q_{1}^{T}A_{1} & Q_{1}^{T}A_{2}\\ Q_{2}^{T}A_{1} & Q_{2}^{T}A_{2} \end{array}\right]\left[\begin{array}{c} X_{1}\\ X_{2} \end{array}\right]=\left[\begin{array}{c} Q_{1}^{T}L\\ Q_{2}^{T}L \end{array}\right] \end{array}$$

According to the mixed total least squares method [54], an augmented matrix is constructed for singular value decomposition to obtain the right singular matrix V, as illustrated in Equation (6).
(6)$$\begin{array}{cc} \left[\begin{array}{cc} Q_{2}^{T}A_{2} & Q_{2}^{T}L \end{array}\right]=U\sum V^{T}, & V=\left[\begin{array}{cc} V_{11} & V_{12}\\ V_{21} & V_{22} \end{array}\right] \end{array}$$

The final deformation parameters obtained are shown in Equation (7).
(7)$$\begin{array}{l} X_{2}=-V_{12}V_{22}^{-1}\\ X_{1}={\left(Q_{1}^{T}A_{1}\right)}^{-1}\left(Q_{1}^{T}L-Q_{1}^{T}A_{2}X_{2}\right) \end{array}$$

### 3.3. Independent Component Analysis for Bridge InSAR Time Series

We employ temporal independent component analysis (tICA) to analyze the spatiotemporal characteristics of deformation using deformation time series data. The tICA technique effectively separates the linear superposition of multiple independent non-Gaussian signals, which have been utilized in InSAR deformation analysis [55,56,57,58,59]. When the number of independent spatiotemporal features, denoted as n, is specified, the FastICA algorithm [60] is applied to estimate the source signal of each independent component, as illustrated in Equation (8),
(8)$$U=\sum_{i}^{n}A_{i}\times S_{i}=\sum_{i}^{n}U_{i}$$
where each row of U represents the deformation time series of one pixel, column represents the number of pixels. The matrix A and S are unknown, which represent the temporal and spatial deformation pattern, respectively, where S*_i_* is the spatial distribution of the *i*th independent source, and A*_i_* is the vector of the relative contribution of the corresponding source S*_i_* [56].

## 4. Experimental Results and Analysis

Firstly, we generated 98 interferograms using a single master SAR image. Subsequently, the temporal coherence is used to identify 15,708 candidate pixels [11] and construct redundant observation networks. To avoid pixel isolation, we set a KNN threshold of 15,708 pixels to create redundant networks for optimizing the phase unwrapping process. Next, we estimated the temporal coherence of each arc, and its statistical histogram is presented in Figure 3A. Meanwhile, a statistical histogram of the temporal coherence derived from Delaunay networks is shown in Figure 3B. It can be observed that after optimizing the redundant networks, numerous high-quality arcs are generated, which enhance the accuracy of the subsequent phase unwrapping.

To maintain high-quality arcs and prevent network isolation, we ultimately retained the network arcs with a temporal coherence threshold greater than 0.85. After removing isolated pixels, we utilized high-quality paths and the remaining 6344 pixels for spatial L1 norm phase unwrapping, thereby obtaining a reliable deformation phase. Figure 4 illustrates the wrapped phase of the 97th interferogram, which represents the final cumulative deformation phase. Distinct interference fringes are clearly visible in this figure. To emphasize the reliability of network optimization-assisted InSAR deformation monitoring, we present the unwrapping phase of the 98th interferogram, as depicted in Figure 5A,B. In these figures, (A) employs the 3D phase unwrapping method proposed by Hooper et al. [61], which has been integrated into the StaMPS software, while (B) utilizes the network optimization method. Figure 5C presents a phase histogram for all 97 unwrapped interferograms. Additionally, Figure 5D displays the cumulative phase time series for point P, as indicated in Figure 5B.

The average standard deviation of the unwrapped phase obtained through the 3D phase unwrapping method is 3.83 radians, whereas the network optimization method yields an average standard deviation of 1.26 radians in Figure 5C. Overall, Figure 5 demonstrates that the network optimization method enhances the reliability of InSAR phase unwrapping, thereby providing robust data for subsequent deformation parameter estimation and analysis.

The SAR data utilized in this article were captured at 10:10:14 UTC. Figure 6 presents the temperature data, in which the blue line represents the highest temperature, the green line indicates the lowest temperature, and the red line reflects the average temperature. These temperature data represent the overall conditions of the Fuzhou region; therefore, they may not accurately represent the temperature directly above the bridge. Lazecky et al. indicate that there is a difference of 11 °C between the bridge road surface temperature and the air temperature [2]. The inaccurate temperature data are used to generate coefficient matrices and calculate the coefficient of thermal expansion. We employed the mixed total least squares method to estimate deformation parameters, thereby mitigating the impact of inaccurate temperature.

Figure 7 illustrates the deformation rate and thermal expansion coefficient estimated by the least squares method for (A) and (B), respectively. Meanwhile, (B) and (C) utilize total least squares, while (D) and (E) employ mixed total least squares. Figure 7A,C,E indicate that no significant deformation was observed in the bridge. In contrast, the thermal expansion coefficient reveals that the cable-stayed bridge deck exhibits notable deformation characteristics, as demonstrated in Figure 7B,D,F.

The total least squares method assumes that all coefficient matrices contain errors. However, time does not contain errors, which introduces additional noise, as illustrated in Figure 7C,D. In contrast, the mixed total least squares method only assumes that the coefficient matrix related to temperature contains errors. Consequently, the deformation parameters obtained from both the least squares method and the mixed total least squares method are similar. The black pentagram serves as the reference point for phase unwrapping.

Points P1 and P2 in Figure 7F correspond to the deformation time series in the vertical direction, as illustrated in Figure 8A,B. The blue line represents the deformation time series, while the green line indicates the temperature deformation time series. The correlation coefficients between the deformation and temperature time series are 0.93 for P1 and −0.91 for P2, respectively.

Moreover, we present various residual phases after the modeling, as shown in Figure 9. Figure 9A–C display the standard deviations of the residual phases corresponding to the least squares method, total least squares method, and mixed total least squares method. Figure 9D illustrates the difference in thermal expansion obtained through the least squares method and the mixed total least squares method. Additionally, the average standard deviation of deformation parameters obtained using the least squares method is 0.68 radians, while the mixed least squares method results in an average standard deviation of 0.64 radians.

## 5. Bridge Deformation Analysis and Discussion

### 5.1. Spatiotemporal Deformation Characteristics

To analyze the spatiotemporal deformation characteristics of Hongtang Bridge, we employed ICA technology to extract four independent components, resulting in four time series evolution features and four spatial features. Figure 10A illustrates the four temporal deformation evolution characteristics, with panel (A) exhibiting distinct periodic deformation patterns similar to those observed in the deformation time series presented in Figure 7B,D,F. In contrast, Figure 10B–D primarily display noise, making the temporal deformation evolution characteristics difficult to discern.

Figure 11A–D present the spatial deformation characteristics corresponding to the temporal deformation evolution characteristics in Figure 10A–D. The deformation magnitude depicted in Figure 11A is constrained to ±20 mm, clearly revealing the deformation characteristics of the bridge expansion joint. Meanwhile, the spatial deformation characteristics shown in Figure 11B–D align with the temporal evolution observed in Figure 10B–D, lacking notable deformation features.

### 5.2. Hongtang Bridge Expansion Joints Deformation Analysis

The previous section discussed the thermal expansion and contraction deformation of bridges. To investigate this phenomenon, we analyzed the original interferometric phase of the bridge at various temperatures. Furthermore, to minimize interferometric fringes caused by DEM errors, we limited the length of the vertical baseline of interferograms. We mapped four wrapped interferograms with smaller perpendicular baselines, as illustrated in Figure 12. The temperature differences (A–D) are 0 °C, −3.5 °C, 8 °C, and 12.5 °C, respectively. The results indicate that as the temperature difference between the two SAR datasets during the generation of interferograms increases, the fringes of the interferometric phase become increasingly dense. This phenomenon is referred to as the thermal expansion and contraction deformation of the bridge. When the temperature difference between the two SAR data acquisition times reaches 12.5 °C, as shown in Figure 12D, a pronounced distribution of the deformation phase is observed along the bridge. This characteristic of thermal expansion is concentrated on both sides of the bridge’s expansion joint, which is particularly relevant to the bridge’s structure.

Figure 13 presents the optical image and photographs from the on-site investigation. In panel (A), the optical image provides a partially zoomed-in view, with the locations of the bridge expansion joints indicated by 11 red lines. Panel (B) features a photograph from the on-site investigation of the bridge expansion joints. Panel (C) displays a photograph of the cable-stayed bridge deck of Hongtang Bridge, located between bridge expansion joints 9 and 10.

In order to better illustrate the deformation characteristics of the bridge, we have constrained the thermal expansion deformation coefficient to a range of 0.6 mm/°C, as depicted in Figure 14B. Panel (A) provides an enlarged partial view that clearly demonstrates the deformation characteristics of the various bridge decks at the interface of the bridge expansion joints.

Figure 14 illustrates the most significant deformation patterns occurring at expansion joints 9 and 10, which are attributed to the cable-stayed bridge deck. The thermal expansion deformation observed on both sides of the cable-stayed bridge deck exhibits a positive correlation with temperature on the left side and a negative correlation on the right side, as shown in Figure 7, with correlation coefficients exceeding +0.9 in (A) and −0.9 in (B), respectively. Thermal expansion occurs in a single direction, leading to opposing deformation patterns on the left and right sides of the cable-stayed bridge deck. The spacing between the cable-stayed bridge decks is 200 m, while the spacing for the other bridge decks is approximately 120 m. The extensive spans of the bridges further amplify these deformations. The large-span bridge deck between sections 9 and 10 necessitates the use of cable-stayed systems for effective stress distribution, which are components of the reinforced steel structures. It also exacerbates the thermal expansion of the cable-stayed bridge deck. 

The deck of Hongtang Bridge in Fuzhou is primarily composed of metal, concrete, and asphalt materials. The physical mechanisms underlying the deformation of bridge expansion joints involve several factors, including temperature fluctuations, concrete shrinkage and creep, and load effects. These factors significantly influence the expansion and deformation of the bridge. The repeated application of vehicle loads contributes to the deformation of the bridge expansion joints, while temperature is the predominant factor affecting the expansion and contraction of the structure [62,63]. Due to temperature fluctuations, significant thermal expansion and contraction can occur, with the extent of deformation varying according to temperature differences. When temperature-induced deformation becomes excessive, it can lead to structural cracking and other issues. The most vulnerable component of the bridge deck is the expansion joint, which is particularly susceptible to damage. Monitoring the deformation of the expansion joint is crucial to ensure that it remains within specified limits, which is vital for the safe operation of bridge structures.

## 6. Conclusions

Bridge expansion joints connect different bridge decks and play a crucial role in releasing loads and accommodating temperature-induced deformations. This study investigated the deformation characteristics of the expansion joints of Hongtang Bridge in Fuzhou, China, using a multi-temporal InSAR technique. We have improved the parameter estimation method for the thermal expansion model of bridges with the mixed total least squares. By integrating optical images and on-site investigations, we observed that temperature-induced significant deformations are most prominent and consistent. Moreover, the thermal expansion deformation is most pronounced at the cable-stayed bridge deck location. Understanding these deformation patterns provides essential insights for evaluating the safety and structural integrity of bridge expansion joints.

## Figures and Tables

**Figure 1 sensors-25-00144-f001:**
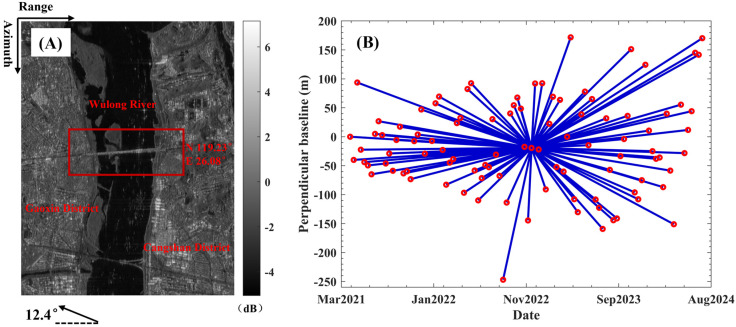
The average SAR intensity image (**A**) and the spatiotemporal baseline of interferograms (**B**).

**Figure 2 sensors-25-00144-f002:**
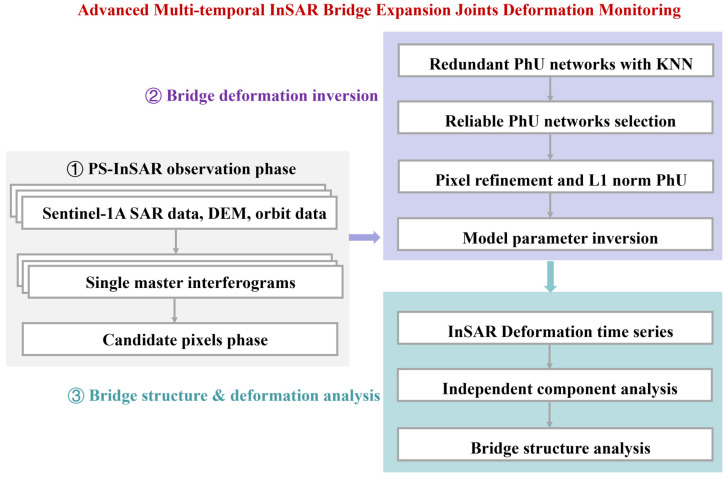
Flowchart for monitoring deformation of bridge expansion joints using MT-InSAR.

**Figure 3 sensors-25-00144-f003:**
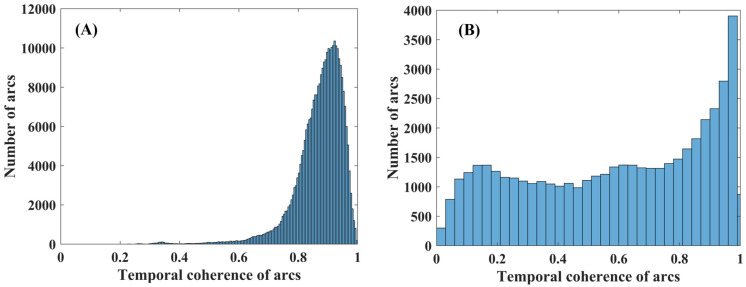
A statistical histogram illustrating the temporal coherence of arcs from optimized networks (**A**) and Delaunay networks (**B**).

**Figure 4 sensors-25-00144-f004:**
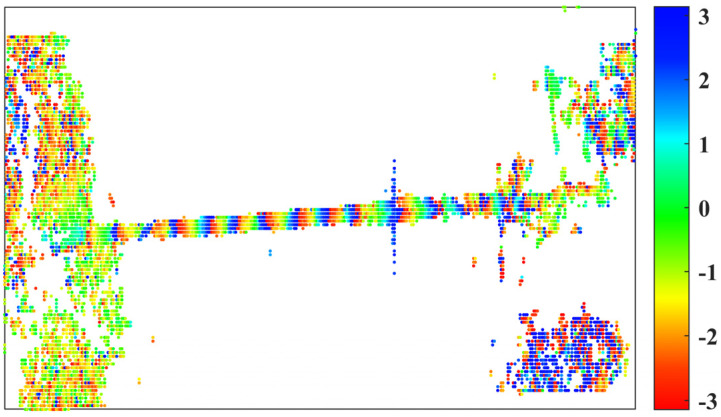
Wrapped phase of the 97-th interferogram.

**Figure 5 sensors-25-00144-f005:**
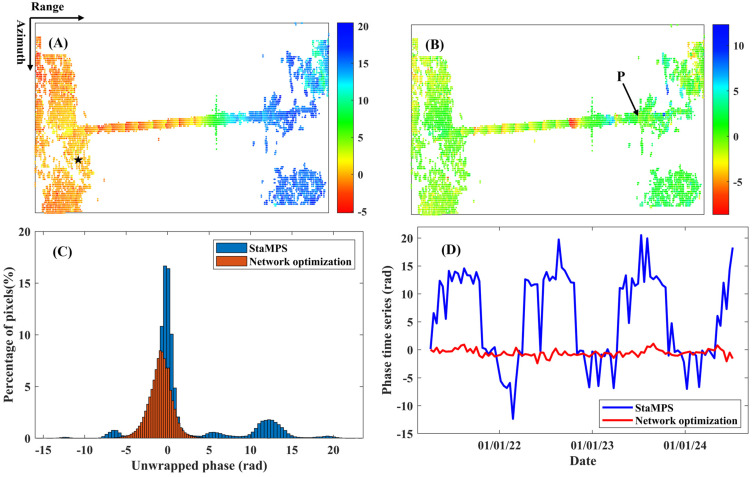
Panels (**A**) and (**B**) illustrate the unwrapped phase using the 3D phase unwrapping method and the network optimization method, respectively. Panel (**C**) presents a histogram of the phase statistics for all unwrapped interferograms, while Panel (**D**) displays the cumulative phase time series for point P, as indicated in Panel (**B**) of Figure 5. The pentagram serves as the reference point for phase unwrapping.

**Figure 6 sensors-25-00144-f006:**
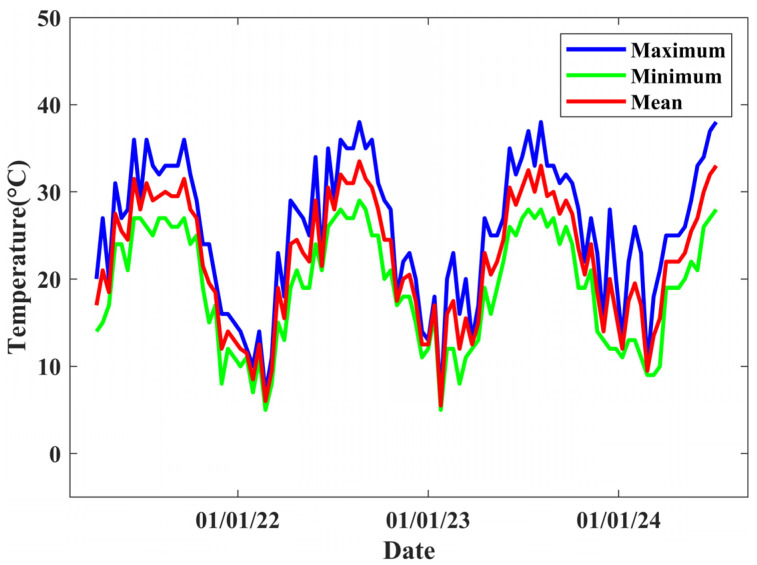
Temperature time series.

**Figure 7 sensors-25-00144-f007:**
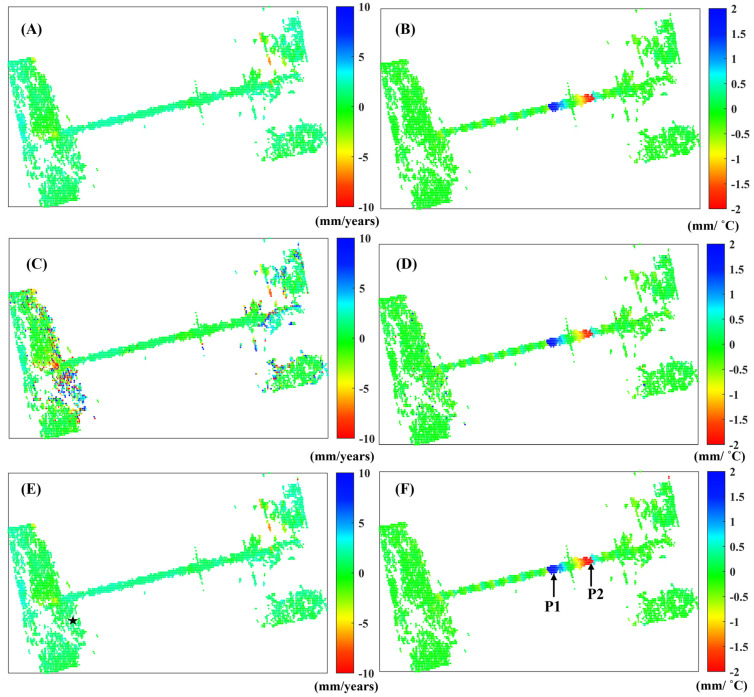
The thermal expansion coefficient and deformation rate were analyzed using different methods. (**A**,**B**) represent the least squares method, (**C**,**D**) denote the total least squares method, while (**E**,**F**) correspond to the mixed total least squares method.

**Figure 8 sensors-25-00144-f008:**
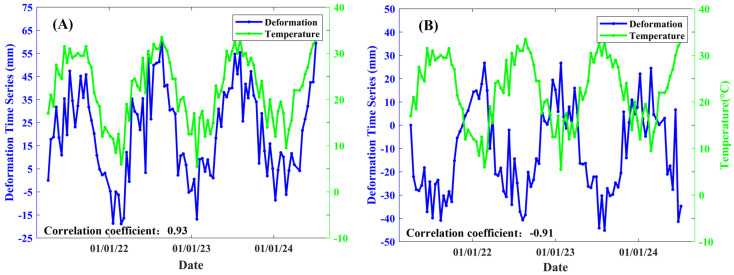
The deformation time series corresponding to P1 and P2, as shown in Figure 7F, are illustrated in panels (**A**,**B**).

**Figure 9 sensors-25-00144-f009:**
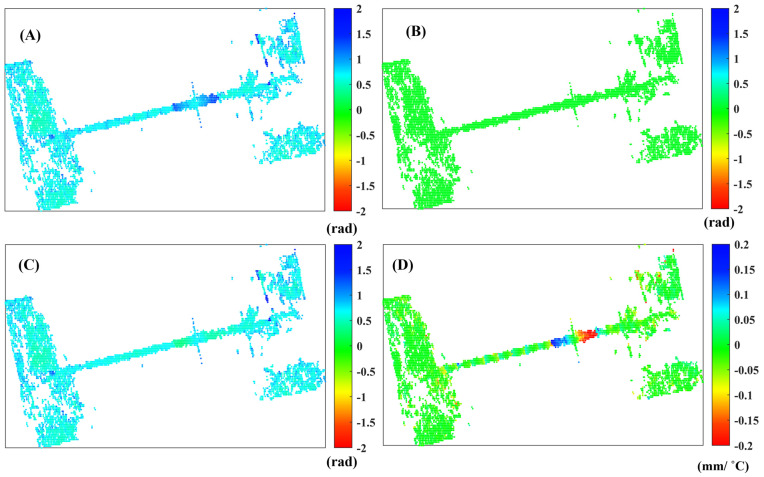
(**A**–**C**) represent the standard deviation of the residual phase after modeling with different methods, while (**D**) indicates the difference between the mixed total least squares method and the least squares method used for estimating the coefficient of thermal expansion.

**Figure 10 sensors-25-00144-f010:**
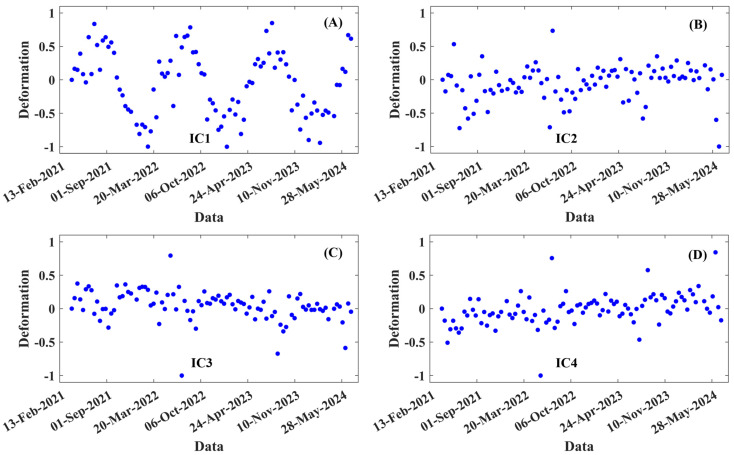
Temporal deformation characteristics of Hongtang Bridge (**A**–**D**) correspond to the independent components IC1 through IC4, respectively.

**Figure 11 sensors-25-00144-f011:**
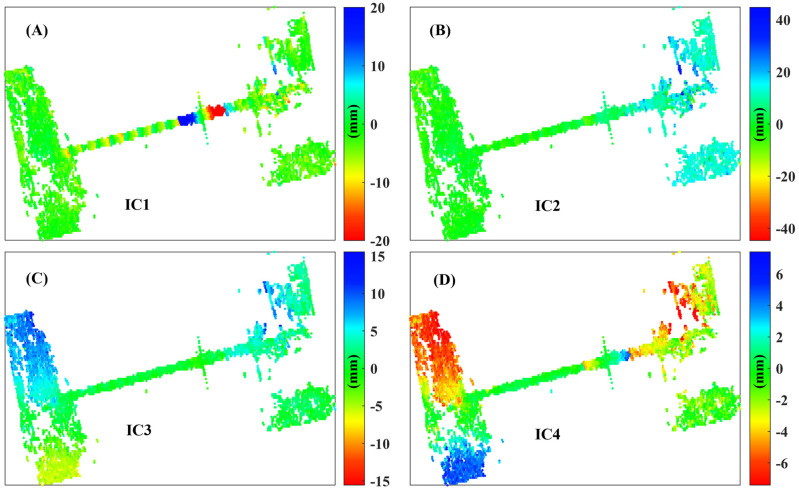
Spatial deformation characteristics of Hongtang Bridge (**A**–**D**) correspond to independent components IC1 through IC4, respectively.

**Figure 12 sensors-25-00144-f012:**
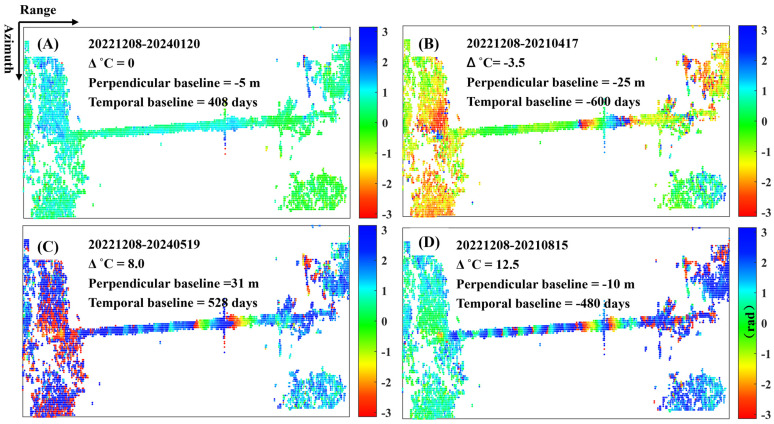
Four wrapped interferograms with varying temperature contrasts.

**Figure 13 sensors-25-00144-f013:**
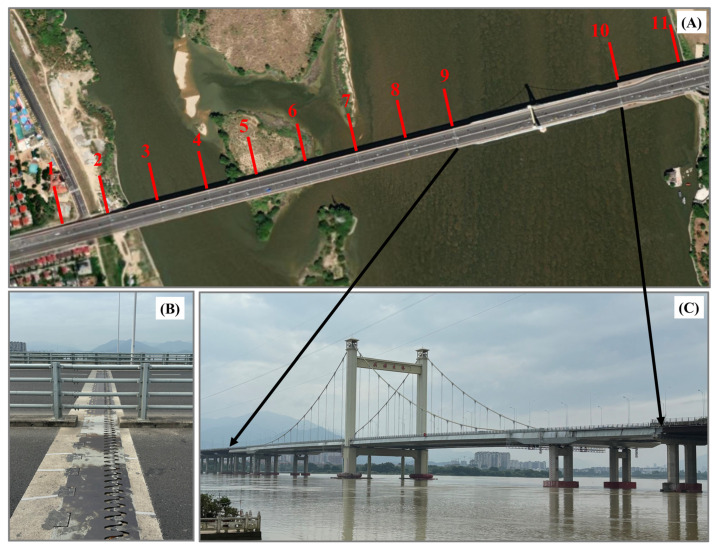
Optical images of Hongtang Bridge (**A**), the expansion joints of the bridge (**B**), and photographs of the cable-stayed bridge deck (**C**). All 11 red lines indicate the locations of the bridge’s expansion joints.

**Figure 14 sensors-25-00144-f014:**
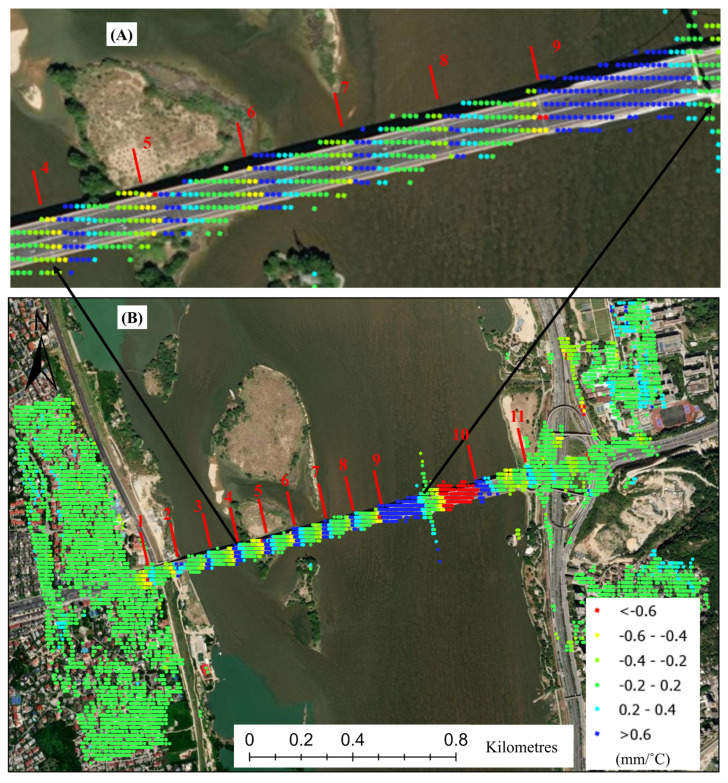
The coefficient of thermal expansion is constrained within the range of 0.6 mm/°C, as overlaid on the optical image (**B**), with a partially enlarged view shown in (**A**).

**Table 1 sensors-25-00144-t001:** Sentinel-1A parameters.

Satellite	Sentinel-1A
Orbit direction	Ascending
Polarization mode	VV
Incidence angle	41.9°
Band	C
Amount	98
Time span	5 April 2021–6 July 2024

## Data Availability

European Space Agency provides the Sentinel-1A data.

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
