# Peer review of "Hongtang Bridge Expansion Joints InSAR Deformation Monitoring with Advanced Phase Unwrapping and Mixed Total Least Squares in Fuzhou China"

_sensors, 2024, doi:10.3390/s25010144_

Round 1

Reviewer 1 Report

Comments and Suggestions for Authors

The paper monitored the deformation characteristics of Fuzhou Hongtang Bridge using InSAR technology. Taking the bridge structure as the research object, the typical deformation locations were identified and their deformation characteristics were evaluated, achieving good research results. However, there are still some shortcomings in the paper. Specific suggestions for improvement are as follows:

1. It is recommended to supplement the information on bridge width, the data source of the DEM, and its resolution.

2. It is suggested to optimize the flowchart in Figure 2, highlighting the unique features and innovations of this paper in the process.

3. In the paper, "A temporal coherence threshold of 0.7 was established to select 15,708 pixels, as illustrated in Figure 3 (A)." It is suggested to include in Figure 3 an additional map depicting the coherence distribution of the temporal interferometric pairs, followed by a map showing the distribution of pixels with coherence values exceeding the 0.7 threshold.

4. In Figure 14(A), the optical image shows that there are approximately 4 monitoring points within the width of any cross-section of the bridge deck. What is the ground resolution of Sentinel-1A? What was the ratio of the number of multi-looks used during data processing? Does the number of pixels within the bridge deck width align with the number of image elements? It is recommended to provide an explanatory analysis. Additionally, the distribution of monitoring points on the bridge deck in the main study area is uniform, and low-coherence points or noise points were basically not filtered out during data processing. Therefore, what is the significance of using network optimization to assist InSAR pixel selection in "Section 3.1" and Figure 3(B)? The authors are asked to provide relevant explanations.

5. The accuracy of the time-series InSAR monitoring results has not been validated.

Author Response

Thank you for your suggestion. The attachment includes our modifications and responses.

Reviewer 2 Report

Comments and Suggestions for Authors

Authors present Hongtang bridge expansion joints deformation monitoring with sentinel-1A interferometry in this manuscript. And there exists 7 major issues. Comments are listed as follows.

Major issue 1: Authors should summarize main contributions of this manuscript in the part of introduction.

Major issue 2: It is better to summarize 1-3 keywords of the proposed method and use these keywords in the title of this manuscript.

Major issue 3: Directions of arrows in Figure 2 may mislead readers. Authors should cancel such misleading.

Major issue 4: Authors should present more detailed description of phase unwrapping in the part of section 3.1.

Major issue 5: Authors should present description on establishing observation network with KNN in the Section 3.

Major issue 6: In figure 5, whats meaning of StaMPS?

Major issue 7: In the part of experiments lacks comparison.

Author Response

(The authors gave the same response as above.)

Reviewer 3 Report

Comments and Suggestions for Authors

The paper investigates “Hongtang Bridge Expansion Joints Deformation Monitoring 2 with Multi-temporal Sentinel-1A Interferometry in Fuzhou 3 Chin”. I have the following comments:

1. Statistical analysis should be conducted to test how significant the results and what confidence level.

2. Abstract shows qualitative superiority of the proposed method over the current state of the art methods.  However, quantitative values should be included in the abstract and the percentage of improvements.

3. Conclusion shows qualitative superiority of the proposed method over the current state of the art methods.  However, quantitative values should be included in the abstract and the percentage of improvements.

4. Comparisons with current state-of-the-art methods should be included to validate the proposed methodology.

5. InSAR is a very well-established methodology for displacement analysis. The authors need to briefly explain their contribution in such a field of research. 

Author Response

(The authors gave the same response as above.)

Reviewer 4 Report

Comments and Suggestions for Authors

In this paper, the authors proposed a method to assess the bridge expansion joints deformation using InSAR. The results is interesting, and the study is valuable. To make the manuscript better, I think the authors should state clearly the kernel of the manuscript. Do the authors prefer to show their new algorithm, or to analyze the deformation of Hongtai Bridge? In the manuscript, I believe that the authors prefer the second one. If so, please explain why do you select Hongtai Bridge? What is the special characteritics make you select this bridge but not others?

Here are some suggestions:

1.Please insert citations of URL using the standard format, but not write the link directly in the main text, i.e., Line 89.

2.In Fig. 1 the authors show the spatio-temporal baselines of the interferograms, a single reference image is selected which is imaged at the center time of your research. However, the deformation results looks like SBAS method have been adopted. Please check it.

3.The (3) just give the expression of L1 norm phase unwrapping theory. It provides no help to understand the manuscript, please remove it.

4.Line 208, I think the authors intended to cite (9) instead of (8).

5.Line 222, why use 0.7 as the threshold? Please add explaination or insert citations.

6.Line 240, the author cited figure 4 after the figure, that is not correct. We need to firstly see the figure citation and explaination in the main text, then secondly the figures themselves. Please adjust the orders.

7.Line 263, mitigating the impact of inaccurate temporature is not the the results caused by the total lease squares estimation method. The method just share the errors within the parameters. Besides, the temperature error is 11°C, which too large to be shared. Please explain it.

8.Figure 7 show a wiered deformation differences. Why the deformation pattern exhibit like this? I think that may be caused by the different movement directions between P1 and P2. Please explain the phenonmena in the manuscript.

9.Please add unit to Fig. 10.

10.Please explain Fig. 12 in the main text with more words.

Author Response

(The authors gave the same response as above.)

Round 2

Reviewer 1 Report

Comments and Suggestions for Authors

The author has made the revisions quite seriously and agrees to accept this paper.

Reviewer 2 Report

Comments and Suggestions for Authors

Authors have answered all issues which reviewer concerns.

Reviewer 3 Report

Comments and Suggestions for Authors

The authors considered my comments